# Manifold GPLVMs for discovering non-Euclidean latent structure in neural data

**Kristopher T. Jensen**
Department of Engineering
University of Cambridge
ktj21@cam.ac.uk

**Ta-Chu Kao**
Department of Engineering
University of Cambridge
tck29@cam.ac.uk

**Marco Tripodi**
Neurobiology Division
MRC Laboratory of Molecular Biology
mtripodi@mrc-lmb.cam.ac.uk

**Guillaume Hennequin**
Department of Engineering
University of Cambridge
g.hennequin@eng.cam.ac.uk

## Abstract

A common problem in neuroscience is to elucidate the collective neural representations of behaviorally important variables such as head direction, spatial location, upcoming movements, or mental spatial transformations. Often, these latent variables are internal constructs not directly accessible to the experimenter. Here, we propose a new probabilistic latent variable model to simultaneously identify the latent state and the way each neuron contributes to its representation in an unsupervised way. In contrast to previous models which assume Euclidean latent spaces, we embrace the fact that latent states often belong to symmetric manifolds such as spheres, tori, or rotation groups of various dimensions. We therefore propose the manifold Gaussian process latent variable model (mGPLVM), where neural responses arise from (i) a shared latent variable living on a specific manifold, and (ii) a set of non-parametric tuning curves determining how each neuron contributes to the representation. Cross-validated comparisons of models with different topologies can be used to distinguish between candidate manifolds, and variational inference enables quantification of uncertainty. We demonstrate the validity of the approach on several synthetic datasets, as well as on calcium recordings from the ellipsoid body of *Drosophila melanogaster* and extracellular recordings from the mouse anterodorsal thalamic nucleus. These circuits are both known to encode head direction, and mGPLVM correctly recovers the ring topology expected from neural populations representing a single angular variable.

## 1 Introduction

The brain uses large neural populations to represent low-dimensional quantities of behavioural relevance such as location in physical or mental spaces, orientation of the body, or motor plans. It is therefore common to project neural data into smaller latent spaces as a first step towards linking neural activity to behaviour (Cunningham and Byron, 2014). This can be done using a variety of linear methods such as PCA or factor analysis (Cunningham and Ghahramani, 2015), or non-linear dimensionality reduction techniques such as tSNE (Maaten and Hinton, 2008). Many of these methods are explicitly probabilistic, with notable examples including GPFA (Yu et al., 2009) and LFADS (Pandarinath et al., 2018). However, all these models project data into Euclidean latent spaces, thus failing to capture the inherent non-Euclidean nature of variables such as head direction or rotational motor plans (Chaudhuri et al., 2019; Finkelstein et al., 2015; Seelig and Jayaraman, 2015; Wilson et al., 2018).

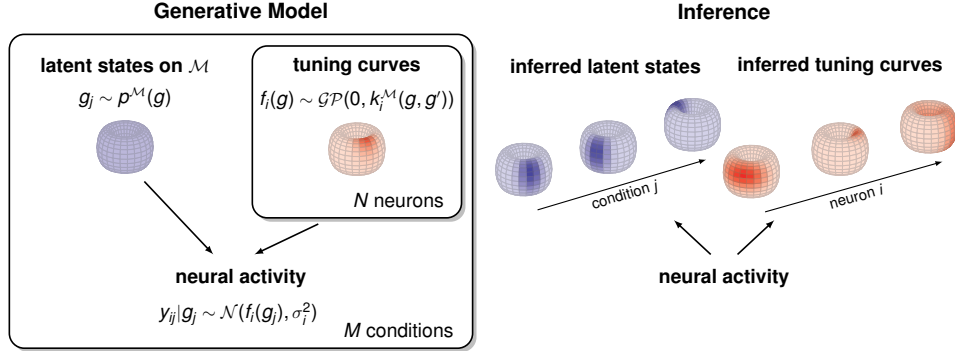

Figure 1: **Schematic illustration of the manifold Gaussian process latent variable model (mGPLVM).** In the generative model (left), neural activity arises from (i) $M$ latent states $\{g_j\}$ on a manifold $\mathcal{M}$, each corresponding to a different condition $j$ (e.g. time or stimulus), and (ii) the tuning curves of $N$ neurons, modelled as Gaussian processes and sharing the same latent states $\{g_j\}$ as inputs. Using variational inference, mGPLVM jointly infers the global latent states and the tuning curve of each neuron on the manifold (right).

Most models in neuroscience justifiably assume that neurons are smoothly tuned (Stringer et al., 2019). As an example, a population of neurons representing an angular variable $\theta$ would respond similarly to some $\theta$ and to $\theta + \epsilon$ (for small $\epsilon$). While it is straigthforward to model such smoothness by introducing smooth priors for response functions defined over $\mathbb{R}$, the activity of neurons modelled this way would exhibit a spurious discontinuity as the latent angle changes from $2\pi$ to $0 + \epsilon$. We see that appropriately modelling smooth neuronal representations requires keeping the latent variables of interest on their natural manifold (here, the circle), instead of an ad-hoc Euclidean space. While periodic kernels have commonly been used to address such problems in GP regression (MacKay, 1998), topological structure has not been incorporated into GP-based latent variable models due to the difficulty of doing inference in such spaces.

Here, we build on recent advances in non-Euclidean variational inference (Falorsi et al., 2019) to develop the manifold Gaussian process latent variable model (mGPLVM), an extension of the GPLVM framework (Lawrence, 2005; Titsias and Lawrence, 2010; Wu et al., 2018, 2017) to non-Euclidean latent spaces including tori, spheres and $SO(3)$ (Figure 1). mGPLVM jointly learns the fluctuations of an underlying latent variable $g$ *and* a probabilistic "tuning curve" $p(f_i|g)$ for each neuron $i$. The model therefore provides a fully unsupervised way of querying how the brain represents its surroundings and a readout of the relevant latent quantities. Importantly, the probabilistic nature of the model enables principled model selection between candidate manifolds. We provide a framework for scalable inference and validate the model on both synthetic and experimental datasets.

## 2 Manifold Gaussian process latent variable model

The main contribution of this paper is mGPLVM, a Gaussian process latent variable model (Titsias and Lawrence, 2010; Wu et al., 2018) defined for non-Euclidean latent spaces. We first present the generative model (Section 2.1), then explain how we perform approximate inference using reparameterizations on Lie groups (Falorsi et al., 2019; Section 2.2). Lie groups include Euclidean vector spaces $\mathbb{R}^n$ as well as other manifolds of interests to neuroscience such as tori $T^n$ (Chaudhuri et al., 2019; Rubin et al., 2019) and the special orthogonal group $SO(3)$ (Finkelstein et al., 2015; Wilson et al., 2018; extensions to non-Lie groups are discussed in Appendix D). We then provide specific forms for variational densities and kernels on tori, spheres, and $SO(3)$ (Section 2.3). Finally we validate the method on both synthetic data (Section 3.1), calcium recordings from the fruit fly head direction system (Section 3.2), and extracellular recordings from the mouse anterodorsal thalamic nucleus (Appendix A).

### 2.1 Generative model

We use $x_{ij}$ to denote the individual elements of a matrix $\boldsymbol{X}$. Let $\boldsymbol{Y} \in \mathbb{R}^{N \times M}$ be the activity of $N$ neurons recorded in each of $M$ conditions. Examples of "conditions" include time within a trial,

stimulus identity, or motor output. We assume that all neuronal responses collectively encode a shared, condition-specific latent variable $g_j \in \mathcal{M}$, where $\mathcal{M}$ is some manifold. We further assume that each neuron $i$ is tuned to the latent state $g$ with a "tuning curve" $f_i(g)$, describing its average response conditioned on $g$. Rather than assuming a specific parametric form for these tuning curves, we place a Gaussian process prior on $f_i(\cdot)$ to capture the heterogeneity widely observed in biological systems (Churchland and Shenoy, 2007; Hardcastle et al., 2017). The model is depicted in Figure 1 and can be formally described as:

$$g_j \sim p^{\mathcal{M}}(g) \qquad\qquad \text{(prior over latents)} \qquad (1)$$

$$f_i \sim \mathcal{GP}(0, k_i^{\mathcal{M}}(\cdot, \cdot)) \qquad\qquad \text{(prior over tuning curves)} \qquad (2)$$

$$y_{ij}|g_j \sim \mathcal{N}(f_i(g_j), \sigma_i^2) \qquad\qquad \text{(noise model)} \qquad (3)$$

In Equation 1, we use a uniform prior $p^{\mathcal{M}}(g)$ inversely proportional to the volume of the manifold for bounded manifolds (Appendix B), and a Gaussian prior on Euclidean spaces to set a basic lengthscale. In Equation 2, $k_i^{\mathcal{M}}(\cdot, \cdot) : \mathcal{M} \times \mathcal{M} \to \mathbb{R}$ is a covariance function defined on manifold $\mathcal{M}$ – manifold-specific details are discussed in Section 2.3. In the special case where $\mathcal{M}$ is a Euclidean space, this model is equivalent to the standard Bayesian GPLVM (Titsias and Lawrence, 2010). While Equation 3 assumes independent noise across neurons, noise correlations can also be introduced as in (Wu et al., 2018) and Poisson noise as in (Wu et al., 2017).

This probabilistic model can be fitted by maximizing the log marginal likelihood

$$\log p(\boldsymbol{Y}) = \log \int p(\boldsymbol{Y}|\{f_i\}, \{g_j\}) \, p(\{f_i\}) \, p^{\mathcal{M}}(\{g_j\}) \, d\{f_i\}d\{g_j\}. \qquad (4)$$

Following optimization, we can query both the posterior over latent states $p(\{g_j\}|\boldsymbol{Y})$ and the posterior predictive distribution $p(\boldsymbol{Y}^\star|\mathcal{G}^\star, \boldsymbol{Y})$ at a set of query states $\mathcal{G}^\star$. While it is possible to marginalise out $f_i$ when the states $\{g_j\}$ are known, further marginalising out $\{g_j\}$ is intractable and maximizing Equation 4 requires approximate inference.

## 2.2 Learning and inference

To maximize $\log p(\boldsymbol{Y})$ in Equation 4, we use variational inference as previously proposed for GPLVMs (Titsias and Lawrence, 2010). The true posterior over the latent states, $p(\{g_j\}|\boldsymbol{Y})$, is approximated by a variational distribution $Q_\theta(\{g_j\})$ with parameters $\theta$ that are optimized to minimize the KL divergence between $Q_\theta(\{g_j\})$ and $p(\{g_j\}|\boldsymbol{Y})$. This is equivalent to maximizing the evidence lower bound (ELBO) on the log marginal likelihood:

$$\mathcal{L}(\theta) = H(Q_\theta) + \mathbb{E}_{Q_\theta}[\log p^{\mathcal{M}}(\{g_j\})] + \mathbb{E}_{Q_\theta}[\log p(\boldsymbol{Y}|\{g_j\})]. \qquad (5)$$

Here, $\mathbb{E}_{Q_\theta}[\cdot]$ indicates averaging over the variational distribution and $H(Q_\theta)$ is its entropy. For simplicity, and because our model does not specify *a priori* statistical dependencies between the individual elements of $\{g_j\}$, we choose a variational distribution $Q_\theta$ that factorizes over conditions:

$$Q_\theta(\{g_j\}) = \prod_{j=1}^{M} q_{\theta_j}(g_j). \qquad (6)$$

In the Euclidean case, the entropy and expectation terms in Equation 5 can be calculated analytically for some kernels (Titsias and Lawrence, 2010), and otherwise using the reparameterization trick (Kingma and Welling, 2014; Rezende et al., 2014). Briefly, the reparameterization trick involves first sampling from a fixed, easy-to-sample distribution (e.g. a normal distribution with zero mean and unit variance), and applying a series of differentiable transformations to obtain samples from $Q_\theta$. We can then use these samples to estimate the entropy term and expectations in Equation 5.

For non-Euclidean manifolds, inference in mGPLVMs poses two major problems. Firstly, we can no longer calculate the ELBO analytically nor evaluate it using the standard reparameterization trick. Secondly, evaluating the Gaussian process log marginal likelihood $\log p(\boldsymbol{Y}|\{g_j\})$ exactly becomes computationally too expensive for large datasets. We address these issues in the following.

### 2.2.1 Reparameterizing distributions on Lie groups

To estimate and optimize the ELBO in Equation 5 when $Q_\theta$ is defined on a non-Euclidean manifold, we use Falorsi et al.'s ReLie framework, an extension of the standard reparameterization trick to variational distributions defined on Lie groups.

**Sampling from $Q_\theta$**   Since we assume that $Q_\theta$ factorizes (Equation 6), sampling from $Q_\theta$ is performed by independently sampling from each $q_{\theta_j}$. We start from a differentiable base distribution $r_{\theta_j}(\boldsymbol{x})$ in $\mathbb{R}^n$. Note that $\mathbb{R}^n$ is isomorphic to the tangent space at the identity element of the group $G$, known as the Lie algebra. We can thus define a 'capitalized' exponential map $\mathrm{Exp}_G : \mathbb{R}^n \to G$, which maps elements of $\mathbb{R}^n$ to elements in $G$ (Sola et al., 2018; Appendix C). Importantly, $\mathrm{Exp}_G$ maps a distribution centered at zero in $\mathbb{R}^n$ to a distribution $\tilde{q}_{\theta_j}$ in the group centered at the identity element. To obtain samples from a distribution $q_{\theta_j}$ centered at an arbitrary $g_j^\mu$ in the group, we can simply apply the group multiplication with $g_j^\mu$ to samples from $\tilde{q}_{\theta_j}$. Therefore, obtaining a sample $g_j$ from $q_{\theta_j}$ involves the following steps: (i) sample from $r_{\theta_j}(\boldsymbol{x})$, (ii) apply $\mathrm{Exp}_G$ to obtain a sample $\tilde{g}_j$ from $\tilde{q}_{\theta_j}$, and (iii) apply the group multiplication $g_j = g_j^\mu \tilde{g}_j$.

**Estimating the entropy $H(Q_\theta)$**   Since $H(q_{\theta_j}) = H(\tilde{q}_{\theta_j})$ (Falorsi et al., 2019), we use $K$ independent Monte Carlo samples from $\tilde{Q}_\theta(\cdot) = \prod_{j=1}^M \tilde{q}_{\theta_j}(\cdot)$ to calculate

$$H(Q_\theta) \approx -\frac{1}{K} \sum_{k=1}^K \sum_{j=1}^M \log \tilde{q}_{\theta_j}(\tilde{g}_{jk}), \tag{7}$$

where $\tilde{g}_{jk} = \mathrm{Exp}_G \boldsymbol{x}_{jk}$ and $\{\boldsymbol{x}_{jk} \sim r_{\theta_j}(\boldsymbol{x})\}_{k=1}^K$.

**Evaluating the density $\tilde{q}_\theta$**   To evaluate $\log \tilde{q}_{\theta_j}(\mathrm{Exp}_G \boldsymbol{x}_{jk})$, we use the result from Falorsi et al. (2019) that

$$\tilde{q}_\theta(\tilde{g}) = \sum_{\boldsymbol{x} \in \mathbb{R}^n \,:\, \mathrm{Exp}_G(\boldsymbol{x}) = \tilde{g}} r_\theta(\boldsymbol{x}) |\boldsymbol{J}(\boldsymbol{x})|^{-1} \tag{8}$$

where $\boldsymbol{J}(\boldsymbol{x})$ is the Jacobian of $\mathrm{Exp}_G$ at $\boldsymbol{x}$. Thus, $\tilde{q}_\theta(\tilde{g})$ is the sum of the Jacobian-weighted densities $r_\theta(\boldsymbol{x})$ in $\mathbb{R}^n$ at *all* those points that are mapped to $\tilde{g}$ through $\mathrm{Exp}_G$ This is an infinite but converging sum, and following Falorsi et al. (2019) we approximate it by its first few dominant terms (Appendix I).

Note that $\mathrm{Exp}_G(\cdot)$ and the group multiplication by $g^\mu$ are both differentiable operations. Therefore, as long as we choose a differentiable base distribution $r_\theta(\boldsymbol{x})$, we can perform end-to-end optimization of the ELBO. In this work we choose the reference distribution to be a multivariate normal $r_{\theta_j}(\boldsymbol{x}) = \mathcal{N}(\boldsymbol{x}; 0, \boldsymbol{\Sigma}_j)$ for each $q_{\theta_j}$. We variationally optimize both $\{\boldsymbol{\Sigma}_j\}$ and the mean parameters $\{g_j^\mu\}$ for all $j$, and together these define the variational distribution.

### 2.2.2   Sparse GP approximation

To efficiently evaluate the $\mathbb{E}_{Q_\theta}[\log p(\boldsymbol{Y}|\{g_j\})]$ term in the ELBO for large datasets, we use the variational sparse GP approximation (Titsias, 2009) which has previously been applied to Euclidean GPLVMs (Titsias and Lawrence, 2010). Specifically, we introduce a set of $m$ inducing points $\mathcal{Z}_i$ for each neuron $i$, and use a lower bound on the GP log marginal likelihood:

$$\log p(\boldsymbol{y}_i|\{g_j\}) \geq \underbrace{-\frac{1}{2}\boldsymbol{y}_i^T(\boldsymbol{Q}_i + \sigma_i^2 \boldsymbol{I})^{-1}\boldsymbol{y}_i - \frac{1}{2}\log|\boldsymbol{Q}_i + \sigma_i^2 \boldsymbol{I}| - \frac{1}{2\sigma^2}\mathrm{Tr}(\boldsymbol{K}_i - \boldsymbol{Q}_i)}_{\log \tilde{p}(\boldsymbol{y}_i|\{g_j\})} + \text{const.} \tag{9}$$

$$\text{with } \boldsymbol{Q}_i = \boldsymbol{K}_{\{g_j\}\mathcal{Z}_i}\boldsymbol{K}_{\mathcal{Z}_i\mathcal{Z}_i}^{-1}\boldsymbol{K}_{\mathcal{Z}_i\{g_j\}} \tag{10}$$

where $\boldsymbol{K}_{\mathcal{AB}}$ denotes the Gram matrix associated with any two input sets $\mathcal{A}$ and $\mathcal{B}$. Note that the latents $\{g_j\}$ are shared across all neurons. In this work we optimize the inducing points on $G$ directly, but they could equivalently be optimized in $\mathbb{R}^n$ and projected onto $G$ via $\mathrm{Exp}_G$.

Using the sparse GP framework, the cost of computing the GP likelihood reduces to $\mathcal{O}(Mm^2)$ for each neuron and Monte Carlo sample. This leads to an overall complexity of $\mathcal{O}(KNMm^2)$ for approximating $\mathbb{E}_{Q_\theta}[\log p(\boldsymbol{Y}|\{g_j\})]$ with $K$ Monte Carlo samples, $N$ neurons, $M$ conditions and $m$ inducing points (see Appendix I for further details on complexity and implementation).

### 2.2.3   Optimization

We are now equipped to optimize the ELBO defined in Equation 5 using Monte Carlo samples drawn from a variational distribution $Q_\theta$ defined on a Lie group $G$. To train the model, we use

Adam (Kingma and Ba, 2014) to perform stochastic gradient descent on the following loss function:

$$\mathcal{L}(\theta) = \frac{1}{K} \sum_{k=1}^{K} \left[ \sum_{j=1}^{M} \left( \log p^{\mathcal{M}}(g_{jk}) - \log \tilde{q}_{\theta_j}(\tilde{g}_{jk}) \right) - \sum_{i}^{N} \log \tilde{p}(\boldsymbol{y}_i | \{g_{jk}\}) \right] \quad (11)$$

where a set of $K$ Monte-Carlo samples $\{\tilde{g}_{jk}\}_{k=1}^{K}$ is drawn at each iteration from $\{\tilde{q}_{\theta_j}\}$ as described in Section 2.2.1. In Equation 11, $g_{jk} = g_j^{\mu} \tilde{g}_{jk}$, where $g_j^{\mu}$ is a group element that is optimized together with all other model parameters. Finally, $\log \tilde{p}(\boldsymbol{y}_i | \{g_j\})$ is the lower bound defined in Equation 9 and $p^{\mathcal{M}}(g_{jk})$ is the prior described in Section 2.1. The inner sums run over conditions $j$ and neurons $i$.

### 2.2.4 Posterior over tuning curves

We approximate the posterior predictive distribution over tuning curves by sampling from the (approximate) posterior over latents. Specifically, for a given neuron $i$ and a set of query states $\mathcal{G}^{\star}$, the posterior predictive over $\boldsymbol{f}_i^{\star}$ is approximated by:

$$p(\boldsymbol{f}_i^{\star} | \boldsymbol{Y}, \mathcal{G}^{\star}) = \frac{1}{K} \sum_{k=1}^{K} p(\boldsymbol{f}_i^{\star} | \mathcal{G}^{\star}, \{\mathcal{G}_k, \boldsymbol{Y}\}) \quad (12)$$

where each $\mathcal{G}_k$ is a set of $M$ latent states (one for each condition in $\boldsymbol{Y}$) independently drawn from the variational posterior $Q_{\theta}(\cdot)$. In Equation 12, each term in the sum is a standard Gaussian process posterior (Rasmussen and Williams, 2006), which we approximate as described above (Section 2.2.2; Appendix E; Titsias, 2009).

### 2.3 Applying mGPLVM to tori, spheres and SO(3)

At this stage, we have yet to define the manifold-specific GP kernels $k^{\mathcal{M}}$ described in Section 2.1. These kernels ought to capture the topology of the latent space and express our prior assumptions that the neuronal tuning curves, defined on the manifold, have certain properties such as smoothness. Here we take inspiration from the common squared exponential covariance function defined over Euclidean spaces and introduce analogous kernels on tori, spheres, and $SO(3)$. This leads to the following general form:

$$k^{\mathcal{M}}(g, g') = \alpha^2 \exp\left( -\frac{d_{\mathcal{M}}(g, g')}{2\ell^2} \right) \qquad g, g' \in \mathcal{M} \quad (13)$$

where $\alpha^2$ is a variance parameter, $\ell$ is a characteristic lengthscale, and $d_{\mathcal{M}}(g, g')$ is a manifold-specific distance function. While squared geodesic distances might be intuitive choices for $d(\cdot, \cdot)$ in Equation 13, they result in positive semi-definite (PSD) kernels only for Euclidean latent spaces (Feragen et al., 2015; Jayasumana et al., 2015). Therefore, we build distance functions that automatically lead to valid covariance functions by observing that (i) dot product kernels are PSD, and (ii) the exponential of a PSD kernel is also PSD. Specifically, we use the following manifold-specific dot product-based distances:

$$d_{R^n}(g, g') = ||g - g'||_2^2 \qquad\qquad g \in \mathbb{R}^n \quad (14)$$

$$d_{S^n}(g, g') = 2(1 - g \cdot g') \qquad\qquad g \in \{\boldsymbol{x} \in \mathbb{R}^{n+1}; \|\boldsymbol{x}\| = 1\} \quad (15)$$

$$d_{T^n}(g, g') = 2 \sum_k (1 - g_k \cdot g_k') \qquad g \in \{(g_1, \cdots, g_n); \forall k: g_k \in \mathbb{R}^2, \|g_k\| = 1\} \quad (16)$$

$$d_{SO(3)}(g, g') = 4 \left[ 1 - (g \cdot g')^2 \right] \qquad\qquad g \in \{\boldsymbol{x} \in \mathbb{R}^4; \|\boldsymbol{x}\| = 1\} \quad (17)$$

where we have slightly abused notation by directly using "$g$" to denote a convenient parameterisation of the group elements which we define on the right of each equation. To build intuition, we note that the distance metric on the torus gives rise to a multivariate von Mises function; the distance metric on the sphere leads to an analogous von Mises Fisher function; and the distance metric on $SO(3)$ is $2(1 - \cos \varphi_{\text{rot}})$ where $\varphi_{\text{rot}}$ is the angle of rotation required to transform $g$ into $g'$. Notably, all these distance functions reduce to the Euclidean squared exponential kernel in the small angle limit. Laplacian (Feragen et al., 2015) and Matérn (Borovitskiy et al., 2020) kernels have previously been proposed for modelling data on Riemannian manifolds, and these can also be incorporated in mGPLVM.

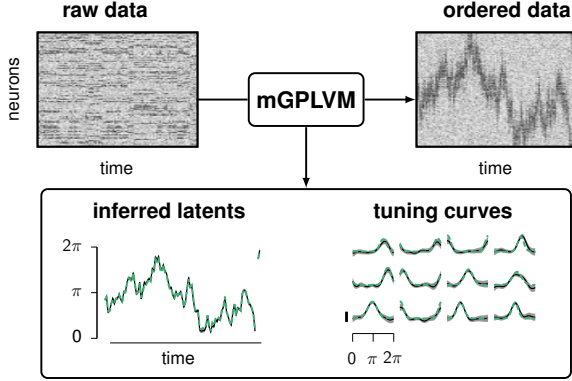

**raw data**     **ordered data**

neurons

time     time

**inferred latents**    **tuning curves**

$2\pi$

$\pi$

$0$

time     $0 \;\; \pi \;\; 2\pi$

Figure 2: **Applying mGPLVM to synthetic data on the ring $T^1$. Top left**: neural activity of 100 neurons at 100 different conditions (here, time bins). **Bottom**: timecourse of the latent states (left) and tuning curves for 12 representative neurons (right). Green: ground truth; Black: posterior mean; Grey shaded regions: $\pm 2$ posterior s.t.d. **Top right**: data replotted from the top left panel, with neurons reordered according to their preferred angles as determined by the inferred tuning curves.

Finally, we provide expressions for the variational densities (Equation 8) defined on tori, $S^3$ and $SO(3)$:

$$\tilde{q}_\theta(\text{Exp}_{T^n}\boldsymbol{x}) = \sum_{\boldsymbol{k}\in\mathbb{Z}^n} r_\theta(\boldsymbol{x} + 2\pi\boldsymbol{k}), \tag{18}$$

$$\tilde{q}_\theta(\text{Exp}_{SO(3)}\boldsymbol{x}) = \sum_{k\in\mathbb{Z}} \left[ r_\theta(\boldsymbol{x} + \pi k\hat{\boldsymbol{x}}) \, \frac{2\|\boldsymbol{x} + \pi k\hat{\boldsymbol{x}}\|^2}{1 - \cos\left(2\|\boldsymbol{x} + \pi k\hat{\boldsymbol{x}}\|\right)} \right], \tag{19}$$

$$\tilde{q}_\theta(\text{Exp}_{S^3}\boldsymbol{x}) = \sum_{k\in\mathbb{Z}} \left[ r_\theta(\boldsymbol{x} + 2\pi k\hat{\boldsymbol{x}}) \, \frac{2\|\boldsymbol{x} + 2\pi k\hat{\boldsymbol{x}}\|^2}{1 - \cos\left(2\|\boldsymbol{x} + 2\pi k\hat{\boldsymbol{x}}\|\right)} \right], \tag{20}$$

where $\hat{\boldsymbol{x}} = \boldsymbol{x}/\|\boldsymbol{x}\|$. Further details and the corresponding exponential maps are given in Appendix C. Since spheres that are not $S^1$ or $S^3$ are not Lie groups, ReLie does not provide a general framework for mGPLVM on these manifolds which we therefore treat separately in Appendix D.

## 3 Experiments and results

In this section, we start by demonstrating the ability of mGPLVM to correctly infer latent states and tuning curves in non-Euclidean spaces using synthetic data generated on $T^1$, $T^2$ and $SO(3)$. We also verify that cross-validated model comparison correctly recovers the topology of the underlying latent space, suggesting that mGPLVM can be used for model selection given a set of candidate manifolds. Finally, we apply mGPLVM to a biological dataset to show that it is robust to the noise and heterogeneity characteristic of experimental recordings.

### 3.1 Synthetic data

To generate synthetic data $\mathbf{Y}$, we specify a target manifold $\mathcal{M}$, draw a set of $M$ latent states $\{g_j\}$ on $\mathcal{M}$, and assign a tuning curve to each neuron $i$ of the form

$$f_i(g) = a_i^2 \exp\left( -\frac{d_{\text{geo}}^2(g, g_i^{\text{pref}})}{2b_i^2} \right) + c_i, \tag{21}$$

$$y_{ij}|g_j \sim \mathcal{N}(f_i(g_j), \sigma_i^2) \tag{22}$$

with random parameters $a_i$, $b_i$ and $c_i$. Thus, the activity of each neuron is a noisy bell-shaped function of the geodesic distance on $\mathcal{M}$ between the momentary latent state $g_j$ and the neuron's preferred state $g_i^{\text{pref}}$ (sampled uniformly). While this choice of tuning curves is inspired by the common 'Gaussian bump' model of neural tuning, we emphasize that the non-parametric prior over $f_i$ in mGPLVM can discover any smooth tuning curve on the manifold, not just Gaussian bumps. For computational simplicity, here we constrain the mGPLVM parameters $\alpha_i$, $\ell_i$ and $\sigma_i$ to be identical across neurons. Note that we can only recover the latent space up to symmetries which preserve pairwise distances. In all figures, we have therefore aligned model predictions and ground truth for ease of visualization (Appendix F).

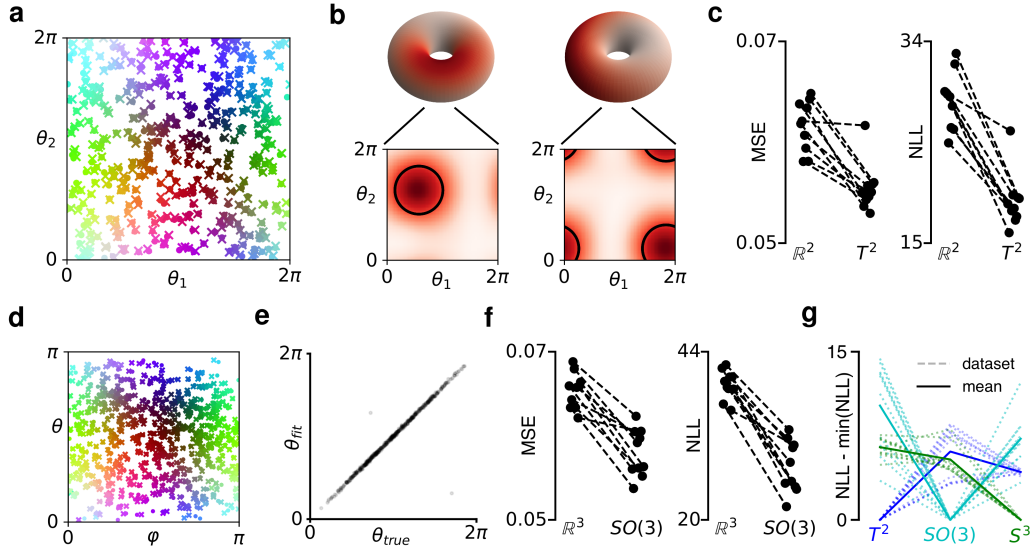

Figure 3: **Validating mGPLVM on synthetic data. (a-c)** Torus dataset. **(a)** True latent states $\{g_j \in T^2\}$ (dots) and posterior latent means $\{g_j^\mu\}$ (crosses). The color scheme is chosen to be smooth for the true latents. **(b)** Posterior tuning curves for two example neurons. Top: tuning curves on the tori. Bottom: projections onto the periodic $[0; 2\pi]$ plane. Black circles indicate locations and widths of the true tuning curves. **(c)** Mean squared cross-validated prediction error (left) and negative log likelihood (right) when fitting $T^2$ and $\mathbb{R}^2$ to data generated on $T^2$. Dashed lines connect datapoints for the same synthetic dataset. **(d-f)** $SO(3)$ dataset. **(d)** Axis of the rotation represented by the true latent states $\{g_j \in SO(3)\}$ (dots) and the posterior latent means $\{g_j^\mu\}$ (crosses) projected onto the $(\varphi, \theta)$-plane. **(e)** Magnitude of the rotations represented by $\{g_j\}$ and $\{g_j^\mu\}$. **(f)** Same as (c), now comparing $SO(3)$ to $\mathbb{R}^3$. **(g)** Test log likelihood ratio for 10 synthetic datasets on $T^2$, $SO(3)$, & $S^3$, with mGPLVM fitted on each manifold (x-axis). Solid lines indicate mean across datasets.

We first generated data on the ring ($T^1$, Figure 2, top left), letting the true latent state be a continuous random walk across conditions for ease of visualization. We then fitted $T^1$-mGPLVM to the data and found that it correctly discovered the true latent states $g$ as well as the ground truth tuning curves (Figure 2, bottom right). Reordering the neurons according to their preferred angles further exposed the population encoding of the angle (Figure 2, top right).

Next, we expanded the latent space to two dimensions with data now populating a 2-torus ($T^2$). Despite the non-trivial topology of this space, $T^2$-mGPLVM provided accurate inference of both latent states (Figure 3a) and tuning curves (Figure 3b). To show that mGPLVM can be used to distinguish between candidate topologies, we compared $T^2$-mGPLVM to a standard Euclidean GPLVM in $\mathbb{R}^2$ on the basis of both cross-validated prediction errors and importance-weighted marginal likelihood estimates (Burda et al., 2015). We simulated 10 different toroidal datasets; for each, we used half the conditions to fit the GP hyperparameters, and half the neurons to predict the latent states for the conditions not used to fit the GP parameters. Finally, we used the inferred GP parameters and latent states to predict the activity of the held-out neurons at the held-out conditions. As expected, the predictions of the toroidal model outperformed those of the standard Euclidean GPLVM which cannot capture the periodic boundary conditions of the torus (Figure 3c).

Beyond toroidal spaces, $SO(3)$ is of particular interest for the study of neural systems encoding 'yaw, pitch and roll' in a variety of 3D rotational contexts (Finkelstein et al., 2015; Shepard and Metzler, 1971; Wilson et al., 2018). We therefore fitted an $SO(3)$-mGPLVM to synthetic data generated on $SO(3)$ and found that it rendered a faithful representation of the latent space and outperformed a Euclidean GPLVM on predictions (Figure 3d-f). Finally we show that mGPLVM can also be used to select between multiple non-Euclidean topologies. We generated 10 datasets on each of $T^2$, $SO(3)$ and $S^3$ and compared cross-validated log likelihoods for $T^2$-, $SO(3)$- and $S^3$-mGPLVM, noting that $p(\mathcal{M}|\boldsymbol{Y}) \propto p(\boldsymbol{Y}|\mathcal{M})$ under a uniform prior over manifolds $\mathcal{M}$. Here we found that the correct latent manifold was consistently the most likely for all 30 datasets (Figure 3g). In summary, these

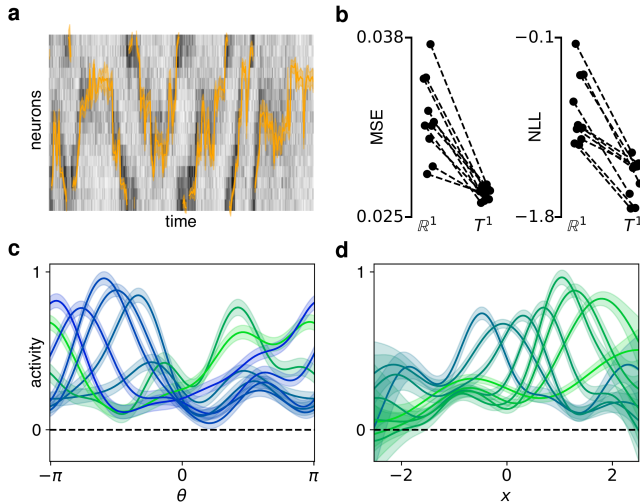

Figure 4: **The *Drosophila* head direction circuit.** **(a)** Input data overlaid with the posterior variational distribution over latent states of a $T^1$-mGPLVM. **(b)** Mean cross-validated prediction error (left) and negative log likelihood (right) for models fitted on $T^1$ and $\mathbb{R}^1$. Each datapoint corresponds to a different partition of the timepoints into a training set and a test set. **(c-d)** Posterior tuning curves for eight example neurons in $T^1$ (c) and $\mathbb{R}^1$ (d). Color encodes the position of the maximum of each tuning curve. Shadings in (a,c,d) indicate $\pm 2$ s.t.d.

results show robust performance of mGPLVM across various manifolds of interest in neuroscience and beyond, as well as a quantitative advantage over Euclidean GPLVMs which ignore the underlying topology of the latent space.

## 3.2 The *Drosophila* head direction circuit

Finally we applied mGPLVM to an experimental dataset to show that it is robust to biological and measurement noise. Here, we used calcium imaging data recorded from the ellipsoid body (EB) of *Drosophila melanogaster* (Turner-Evans, 2020; Turner-Evans et al., 2020), where the so-called E-PG neurons have recently been shown to encode head direction (Seelig and Jayaraman, 2015). The EB is divided into 16 'wedges', each containing 2-3 E-PG neurons that are not distinguishable on the basis of calcium imaging data, and we therefore treat each wedge as one 'neuron'. Due to the physical shape of the EB, neurons come 'pre-ordered' since their joint activity resembles a bump rotating on a ring (Figure 4a, analogous to Figure 2, "ordered data"). While the EB's apparent ring topology obviates the need for mGPLVM as an explorative tool for uncovering manifold representations, we emphasize that head direction circuits in higher organisms are not so obviously structured (Chaudhuri et al. (2019); Appendix A) – in fact, some brain areas such as the entorhinal cortex even embed concurrent representations of multiple spaces (Constantinescu et al., 2016; Hafting et al., 2005).

We fitted the full mGPLVM with a separate GP for each neuron and found that $T^1$-mGPLVM performed better than $\mathbb{R}^1$-mGPLVM on both cross-validated prediction errors and log marginal likelihoods (Figure 4b). The model recovered latent angles that faithfully captured the visible rotation of the activity bump around the EB, with larger uncertainty during periods where the neurons were less active (Figure 4a, orange). When querying the posterior tuning curves from a fit in $\mathbb{R}^1$, these were found to suffer from spurious boundary conditions with inflated uncertainty at the edges of the latent representation – regions where $\mathbb{R}^1$-mGPLVM effectively has less data than $T^1$-mGPLVM since $\mathbb{R}^1$ does not wrap around. In comparison, the tuning curves were more uniform across angles in $T^1$ which correctly captures the continuity of the underlying manifold. In Appendix A, we describe similar results with mGPLVM applied to a dataset from the mouse head-direction circuit with more heterogeneous neuronal tuning and no obvious anatomical organization (Peyrache et al., 2015).

## 4 Discussion and future work

**Conclusion** We have presented an extension of the popular GPLVM model to incorporate non-Euclidean latent spaces. This is achieved by combining a Bayesian GPLVM with recently developed methods for approximate inference in non-Euclidean spaces and a new family of manifold-specific kernels. Inference is performed using variational sparse GPs for computational tractability with inducing points optimized directly on the manifold. We demonstrated that mGPLVM correctly infers the latent states and GP parameters for synthetic data of various dimensions and topologies, and that cross-validated model comparisons can recover the correct topology of the space. Finally, we showed

how mGPLVM can be used to infer latent topologies and representations in biological circuits from calcium imaging data. We expect mGPLVM to be particularly valuable to the neuroscience community because many quantities encoded in the brain naturally live in non-Euclidean spaces (Chaudhuri et al., 2019; Finkelstein et al., 2015; Wilson et al., 2018).

**Related work**  GP-based latent variable models with periodicity in the latent space have previously been used for motion capture, tracking and animation (Elgammal and Lee, 2008; Urtasun et al., 2008). However, these approaches are not easily generalized to other non-Euclidean topologies and do not provide a tractable marginal likelihood which forms the basis of our Bayesian model comparisons. Additionally, methods have been developed for analysing the geometry of the latent space of GPLVMs (Tosi et al., 2014) and other latent variable models (Arvanitidis et al., 2017) after initially learning the models with a Euclidean latent. These approaches confer a degree of interpretability to the learned latent space but do not explicitly incorporate priors and topological constraints on the manifold during learning. Furthermore, GPs and GPLVMs with non-Euclidean outputs have been developed (Mallasto and Feragen, 2018; Mallasto et al., 2019; Navarro et al., 2017). These approaches are orthogonal to mGPLVM where the latent GP inputs, not outputs, live on a non-Euclidean manifold. mGPLVM can potentially be combined with these approaches to model non-Euclidean observations, and to incorporate more expressive GP priors over the latent states than the independent prior we have used here.

Finally, several methods for inference in non-Euclidean spaces have been developed in the machine learning literature. These have centered around methods based on VAEs (Davidson et al., 2018; Rey et al., 2019; Wang and Wang, 2019), normalizing flows (Rezende et al., 2020), and neural ODEs (Falorsi and Forré, 2020; Lou et al., 2020; Mathieu and Nickel, 2020). While non-Euclidean VAEs are useful for amortized inference, they constrain $f(g)$ more than a GP does and do not naturally allow expression of a prior over its smoothness. Normalizing flows and neural ODEs can potentially be combined with mGPLVM to increase the expressiveness of the variational distributions (Falorsi et al., 2019). This would allow us to model complex distributions over latents, such as the multimodal distributions that naturally arise in ambiguous environments with symmetries (Jacob et al., 2017).

**mGPLVM extensions**  Here, we have assumed statistical independence across latent states, but prior dependencies could be introduced to incorporate e.g. temporal smoothness by placing a GP prior on the latents as in GPFA (Yu et al., 2009). To capture more statistical structure in the latents, richer variational approximations of the posterior could be learned by using normalizing flows on the base distribution ($r_\theta$). It would also be interesting to exploit automatic relevance determination (ARD, Neal, 2012) in mGPLVM to automatically select the latent manifold dimension. We explored this approach by fitting a $T^2$-mGPLVM to the data from Figure 2 with separate lengthscales for the two dimensions, where we found that $T^2$ shrunk to $T^1$, the true underlying manifold (Appendix G).

Furthermore, the mGPLVM framework can be extended to direct products of manifolds, enabling the study of brain areas encoding non-Euclidean variables such as head direction jointly with global modulation parameters such as attention or velocity. As an example, fitting a $(T^1 \times \mathbb{R}^1)$-mGPLVM to the *Drosophila* data captures both the angular heading in the $T^1$ dimension as well as a variable correlated with global activity in the $\mathbb{R}^1$ dimension (Appendix H).

**Future applications**  mGPLVM not only infers the most likely latent states but also estimates the associated uncertainty, which can be used as a proxy for the degree of momentary coherence expressed in neural representations. It would be interesting to compare such posterior uncertainties and tuning properties in animals across brain states. For example, uncertainty estimates could be compared across sleep and wakefulness or environments with reliable and noisy spatial cues.

In the motor domain, mGPLVM can help elucidate the neural encoding of motor plans for movements naturally specified in rotational spaces. Examples include 3-dimensional head rotations represented in the rodent superior colliculus (Masullo et al., 2019; Wilson et al., 2018) as well as analogous circuits in primates. Finally, it will be interesting to apply mGPLVM to artificial agents trained on tasks that require them to form internal representations of non-Euclidean environmental variables (Banino et al., 2018). Our framework could be used to dissect such representations, adding to a growing toolbox for the analysis of artificial neural networks (Sussillo and Barak, 2013).

## Acknowledgements

We thank Daniel Turner-Evans and Vivek Jayaraman for sharing their experimental data. We are grateful for helpful comments on the manuscript by Robert Pinsler, Marine Schimel, David Liu, and others in the CBL.

## Broader Impact

There are two broad fields which we expect might be influenced by our work. From a technical point of view, mGPLVM extends a probabilistic machine learning toolbox which has downstream applications in fields ranging from speech recognition and image classification to personalized medicine (Ghahramani, 2015). However, this work is primarily geared towards neuroscience, and that is where we expect it to have the largest impact. One tangible application of methods for probabilistic inference in neural populations is in the fields of brain-machine interfaces (BMI) and neuroprosthetics – methods which allow the brain to control external devices directly and without intermediate motor output. The control of such actuated systems might be more effectively performed by representing high-level plans – such as 3D motor actions, navigational plans or rotation of objects – directly on the relevant non-Euclidean manifolds. We therefore expect that new inference methods in such spaces might accelerate the development of BMIs, fostering a range of medical applications, from amputees using neuroprosthetic devices as substitutes for missing limbs to surgeons operating remote high-precision surgical devices using neural activity directly. These applications come with various ethical and societal concerns; in particular, the ability to automatically extract internal brain representations comes with concerns of privacy. Fortunately, many of these challenges are already being considered by the community and actively explored in the field of bioethics (Clausen, 2009), and we hope that such ethical considerations will continue to shape the way we do research in the future.

## Funding disclosure

K.T.J. was funded by a Gates Cambridge scholarship; T-C.K. by a Trinity-Henry Barlow scholarship and a scholarship from the Ministry of Education, ROC Taiwan; and M.T. by the Medical Research Council (MC_UP_12012) and an ERC Starting Grant (STG 677029).

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
