[Supplementary Material]

# Appendix – Manifold GPLVMs for discovering non-Euclidean latent structure in neural data

## A  The mouse head direction circuit

Figure 5: **The mouse head direction circuit. (a)** Population activity recorded from mouse ADn during foraging. **(b)** Variational mean inferred by $T^1$-mGPLVM plotted against the true mouse head direction. **(c)** Kernel length scales for the 29 neurons recorded. Dashed line: $\ell^2 = 4$ (maximum $d$ in the $T^1$-kernel). Insets: example neurons with low and high $\ell$. **(d)** Tuning curves for three example neurons inferred during wake (black) and REM sleep (red).

To highlight the importance of unsupervised non-Euclidean learning methods in neuroscience and to illustrate the interpretability of the learned GP parameters, we consider a dataset from Peyrache et al. (2015b) recorded from the mouse anterodorsal thalamic nucleus (ADn; Figure 5a). This data has also been analyzed in Peyrache et al. (2015a), Chaudhuri et al. (2019) and Rubin et al. (2019). We consider the same example session shown in Figure 2 of Chaudhuri et al. (2019) (Mouse 28, session 140313) and bin spike counts in 500 ms time bins for analysis with mGPLVM. When comparing cross-validated log likelihoods for $T^1$- and $\mathbb{R}^1$-mGPLVM fitted to the data, $T^1$ consistently outperformed $\mathbb{R}^1$ with a log likelihood ratio of $127 \pm 30$ (mean $\pm$ sem) across 10 partitions of the data.

Fitting $T^1$-mGPLVM to the binned spike data, we found that the inferred latent state was highly correlated with the true head direction (Figure 5b). However, in contrast to the data considered in Section 3.1 and Section 3.2, this mouse dataset contains neurons with more heterogeneous baseline activities and tuning properties. This is reflected in the learned GP parameters which converge to small kernel length scales for neurons that contribute to the heading representation (Figure 5c, 'tuned') and large length scales for those that do not (Figure 5c, 'not tuned'). Finally, since mGPLVM does not require knowledge of behaviour, we also fitted mGPLVM to data recorded from the same neurons during a period of rapid eye movement (REM) sleep. Here we found that the representation of subconscious heading during REM sleep was similar to the representation of heading when the animal was awake after matching the offset between the two sets of tuning curves (Figure 5d), similar to results by Peyrache et al. (2015a). However, their analyses relied on recordings from two separate brain regions to align the activity from neurons in ADn to a subconscious head direction decoded from the postsubiculum and vice versa. In contrast, mGPLVM allows for fully unsupervised Bayesian analyses across both wake and sleep using recordings from a single brain area.

## B  Priors on manifolds

For all manifolds, we use priors that factorize over conditions, $p^{\mathcal{M}}(\{g_j\}) = \prod_j p^{\mathcal{M}}(g_j)$. As described in Section 2.1, we use a Gaussian prior $p^{R^n}(g) = \mathcal{N}(g; 0, \mathbf{I}_n)$ over latent states in $\mathbb{R}^n$, and uniform priors for the spheres, tori, and $SO(3)$. These uniform priors have a density which is the

inverse volume of the manifold:

$$p^{S^n}(g) = \left[\frac{2\pi^{\frac{n+1}{2}}}{\Gamma(\frac{n+1}{2})}\right]^{-1} \tag{23}$$

$$p^{T^n}(g) = [2\pi]^{-n} \tag{24}$$

$$p^{SO(3)}(g) = \left[\frac{2\pi^{\frac{4}{2}}}{2\Gamma(\frac{4}{2})}\right]^{-1}. \tag{25}$$

Note that the volume of $S^n$ is the surface area of the $n$-sphere, and the volume of $SO(3)$ is half the volume of $S^3$.

## C  Lie groups and their exponential maps

For simplicity of exposition, we have skimmed over the details of how the 'capitalized' Exponential map $\text{Exp}_G : \mathbb{R}^n \to G$ is defined in Section 2.2.1, particularly in relation to the group's Lie algebra $\mathfrak{g}$. Here we make this connection more explicit. As described in the main text, the Lie algebra $\mathfrak{g}$ of a group $G$ is a vector space tangent to $G$ at its identity element. The exponential map $\exp_G : \mathfrak{g} \to G$ maps elements from the Lie algebra to the group, and is conceptually distinct from the "capitalised" Exponential map defined in Section 2.2.1 which maps from $\mathbb{R}^n$ to $G$. However, because the Lie algebra is isomorphic to $\mathbb{R}^n$, we have found it convenient in both our exposition and our implementation to work directly with the pair $(\mathbb{R}^n, \text{Exp}_G)$, instead of $(\mathfrak{g}, \exp_G)$. To expand on the connection between the two, note that we can define as in Sola et al. (2018) the isomorphism Hat : $\mathbb{R}^n \to \mathfrak{g}$, which maps every element in $\mathbb{R}^n$ to a distinct element in the Lie algebra $\mathfrak{g}$. Therefore, $\text{Exp}_G : \mathbb{R}^n \to G$ is in fact the composition $\exp_G \circ \text{Hat}$.

### Manifold-specific parameterizations

Here we provide some further justification for the forms of $\tilde{q}_\theta(\tilde{g})$ provided in Equations 18 and 19 as well as the exponential maps which are used to derive these densities and are needed for optimization in Equation 11. For both $T^n$ and $SO(3)$, we use Equation 8 from Falorsi et al. (2019), which we repeat here for reference:

$$\tilde{q}_\theta(\tilde{g}) = \sum_{\boldsymbol{x} \in \mathbb{R}^n \, : \, \text{Exp}_G(\boldsymbol{x}) = \tilde{g}} r_\theta(\boldsymbol{x}) |\boldsymbol{J}(\boldsymbol{x})|^{-1}. \tag{26}$$

In what follows, we will use $\boldsymbol{g}$ to indicate a vector representation of group element $g$ to avoid conflicts of notation.

Note that the expressions in this section largely follow Falorsi et al. (2019), but we re-write them in a different basis for ease of computational implementation.

### C.1  $T^n$

The $n$-Torus $T^n$ is the direct product of $n$ circles, such that we can parameterize members of this group as $\boldsymbol{g} \in \mathbb{R}^n$ whose elements are all angles between 0 and $2\pi$. Note that this is equivalent to the parameterization in Equation 16 except that here we denote an element on the circle by its angle, while in Equation 16 we denote it by a unit 2-vector for notational consistency with the other kernels. Because 1-dimensional rotations are commutative, the parameterization of the torus as a list of angles allows us to perform group operations by simple addition modulo $2\pi$. We therefore slightly abuse notation and write the exponential map $\text{Exp}_{T^n} : \mathbb{R}^n \to T^n$ as an element-wise modulo operation:

$$\text{Exp}_{T^n} \boldsymbol{x} = \boldsymbol{x} \bmod 2\pi. \tag{27}$$

Equation 27 has inverse Jacobian $|\boldsymbol{J}(x)|^{-1} = 1$. Moreover, since $\text{Exp}_{T^n}(\boldsymbol{x}) = \text{Exp}_{T^n}(\boldsymbol{x} + 2\pi\boldsymbol{k})$ for any integer vector $\boldsymbol{k} \in \mathbb{Z}^n$, the change-of-variable formula in Equation 26 yields the following density on $T^n$:

$$\tilde{q}_\theta(\text{Exp}_{T^n} \boldsymbol{x}) = \sum_{\boldsymbol{k} \in \mathbb{Z}^n} r_\theta(\boldsymbol{x} + 2\pi\boldsymbol{k}). \tag{28}$$

For ease of implementation it is also convenient to rewrite the kernel distance function Equation 16 as

$$d_{T^n}(\boldsymbol{g}, \boldsymbol{g}') = 2 \cdot \mathbf{1}_n \cdot (1 - \cos(\boldsymbol{g} - \boldsymbol{g}')) \tag{29}$$

where $\mathbf{1}_n$ is the n-vector full of ones, and $\cos(\cdot)$ is applied element-wise to $\boldsymbol{g} - \boldsymbol{g}'$.

## C.2  $SO(3)$

We use quaternions $\boldsymbol{g} \in \mathbb{R}^4$ to represent elements $g \in SO(3)$ as indicated in Equation 17. For a rotation of $\phi$ radians around axis $\boldsymbol{u} \in \mathbb{R}^3$ with $\|\boldsymbol{u}\| = 1$,

$$\boldsymbol{g} = \left( \cos \frac{\phi}{2}, \boldsymbol{u} \sin \frac{\phi}{2} \right) \in \mathbb{R}^4. \tag{30}$$

The exponential map $\mathrm{Exp}_{SO(3)} : \mathbb{R}^3 \to SO(3)$ is

$$\mathrm{Exp}_{SO(3)} \boldsymbol{x} = (\cos \|\boldsymbol{x}\|, \hat{\boldsymbol{x}} \sin \|\boldsymbol{x}\|), \tag{31}$$

where $\hat{\boldsymbol{x}} = \boldsymbol{x}/\|\boldsymbol{x}\|$ and $\phi = 2\|\boldsymbol{x}\|$ is the angle of rotation. This gives rise to an inverse Jacobian

$$|\boldsymbol{J}(\boldsymbol{x})|^{-1} = \phi^2/(2(1 - \cos \phi)). \tag{32}$$

Using Equation 26 we get the density on the group

$$\tilde{q}_\theta(\mathrm{Exp}_{SO(3)} \boldsymbol{x}) = \sum_{k \in \mathbb{Z}} \left[ r_\theta(\boldsymbol{x} + \pi k \hat{\boldsymbol{x}}) \frac{2\|\boldsymbol{x} + \pi k \hat{\boldsymbol{x}}\|^2}{1 - \cos(2\|\boldsymbol{x} + \pi k \hat{\boldsymbol{x}}\|)} \right], \tag{33}$$

where the sum over $k$ stems from the fact that a rotation of $\phi + 2k\pi$ around axis $\hat{\boldsymbol{x}}$ is equivalent to a rotation of $\phi$ around the same axis.

# D  mGPLVM on $S^n$

In this section, we discuss how to fit mGPLVMs on spheres. We first consider spheres which are also Lie groups, and then discuss a general framework for all $n$-spheres.

## D.1  $S^{1,3}$

We begin by noting that $S^n$ is not a Lie group unless $n = 1$ or $n = 3$, thus we can only apply the ReLie framework to $S^1$ and $S^3$. $S^1$ is equivalent to $T^1$ and is most easily treated using the torus formalism above. For $S^3$, we note that $SO(3)$ is simply $S^3$ with double coverage. This is because quaternions $\boldsymbol{g}$ and $-\boldsymbol{g}$ represent the same element of $SO(3)$ while they correspond to distinct elements of $S^3$. The Jacobian and exponential maps of $S^3$ are therefore identical to those of $SO(3)$. The expression for the density on $S^3$ also mirrors Equation 33 except that the sum is over $\boldsymbol{x} + 2\pi k \hat{\boldsymbol{x}}$ instead of $\boldsymbol{x} + \pi k \hat{\boldsymbol{x}}$:

$$\tilde{q}_\theta(\mathrm{Exp}_{S^3} \boldsymbol{x}) = \sum_{k \in \mathbb{Z}} \left[ r_\theta(\boldsymbol{x} + 2\pi k \hat{\boldsymbol{x}}) \frac{2\|\boldsymbol{x} + 2\pi k \hat{\boldsymbol{x}}\|^2}{1 - \cos(2\|\boldsymbol{x} + 2\pi k \hat{\boldsymbol{x}}\|)} \right]. \tag{34}$$

We demonstrate $S^3$-mGPLVM on synthetic data from $S^3$ in Figure 6 (bottom).

## D.2  $S^{n \notin \{1,3\}}$

The ReLie framework does not directly apply to distributions defined on non-Lie groups. Nevertheless, we can still apply mGPLVM to an $n$-sphere embedded in $\mathbb{R}^{n+1}$ by taking each latent variational distribution $q_{\theta_j}$ to be a von Mises-Fisher distribution (VMF), whose entropy is known analytically. Parameterizing group element $g \in S^n$ by a unit-norm vector $\boldsymbol{g} \in \mathbb{R}^{n+1}$, $\|\boldsymbol{g}\| = 1$, this density is given by:

$$q_\theta(\boldsymbol{g}; \boldsymbol{g}^\mu, \kappa) = \frac{\kappa^{n/2-1}}{(2\pi)^{n/2} I_{n/2-1}(\kappa)} \exp(\kappa \, \boldsymbol{g}^\mu \cdot \boldsymbol{g}) \tag{35}$$

Figure 6: **Applying mGPLVM to synthetic data on $S^2$ (top) and $S^3$ (bottom).** Pairwise distances between the variational means $\{g_j^\mu\}$ are plotted against the corresponding pairwise distances between the true latent states $\{g_j\}$ for $S^2$ (top left) and $S^3$ (bottom left). Since the log likelihood is a function of these pairwise distances through the kernel (Equation 15), this illustrates that mGPLVM recovers the important features of the true latents. Inferred (black) and true (green) latent states in spherical coordinates for $S^2$ (top middle) and $S^3$ (bottom middle and bottom right). For $S^2$, we are showing the latent states in spherical polar coordinates $\boldsymbol{g} = (\sin\theta\cos\varphi, \sin\theta\sin\varphi, \cos\theta)$ with $\theta \in [0, \pi]$ and $\varphi \in [0, 2\pi]$. For $S^3$, we use hyperspherical coordinates $\boldsymbol{g} = (\sin\psi\sin\theta\cos\varphi, \sin\psi\sin\theta\sin\varphi, \sin\theta\cos\psi, \cos\theta)$ with $\theta, \psi \in [0, \pi]$ and $\varphi \in [0, 2\pi]$.

where $\cdot$ denotes the dot product. Here, $I_v$ is the modified Bessel function of the first kind at order $v$, $\boldsymbol{g}^\mu$ is the mean direction of the distribution on the hypersphere, and $\kappa \geq 0$ is a concentration parameter – the larger $\kappa$, the more concentrated the distribution around $\boldsymbol{g}^\mu$.

Using a VMF distribution as the latent distribution, we can easily evaluate the ELBO in Equation 5 because (i) there are well-known algorithms for sampling from the distribution using rejection-sampling (Ulrich, 1984) and (ii) both the entropy term $H(q_\theta)$ and its gradient can be derived analytically (Davidson et al., 2018). For details of how to differentiate through rejection sampling, please refer to Naesseth et al. (2016) and Davidson et al. (2018).

In the following, we provide details for applying mGPLVM to $S^2$ for which we do not need to use rejection sampling and instead use inverse transform sampling (Jakob, 2012). For $S^2$, the VMF distribution simplifies to (Straub, 2017)

$$q_\theta(\boldsymbol{g}; \boldsymbol{g}^\mu, \kappa) = \frac{\kappa}{2\pi(\exp(\kappa) - \exp(-\kappa))} \exp(\kappa\,\boldsymbol{g}^\mu \cdot \boldsymbol{g}), \tag{36}$$

and its entropy is

$$H(q_\theta) = -\int_{S^2} q_\theta(\boldsymbol{g}; \boldsymbol{g}^\mu, \kappa) \log q_\theta(\boldsymbol{g}; \boldsymbol{g}^\mu, \kappa) d\boldsymbol{g} \tag{37}$$

$$= -\log\left(\frac{\kappa}{4\pi\sinh\kappa}\right) - \frac{\kappa}{\tanh\kappa} + 1. \tag{38}$$

These equations allow us to apply mGPLVM to $S^2$ by optimizing the ELBO as described in the main text; this is illustrated for synthetic data on $S^2$ in Figure 6 (top).

## E  Posterior over tuning curves

We can derive the posterior over tuning curves in Equation 12 as follows:

$$p(\boldsymbol{f}_i^\star|\boldsymbol{Y},\mathcal{G}^\star) = \int p(\boldsymbol{f}_i^\star,\mathcal{G}|\mathcal{G}^\star,\boldsymbol{Y})\,d\mathcal{G} \tag{39}$$

$$= \int p(\boldsymbol{f}_i^\star|\mathcal{G}^\star,\{\mathcal{G},\boldsymbol{Y}\})p(\mathcal{G}|\boldsymbol{Y})\,d\mathcal{G} \tag{40}$$

$$\approx \int p(\boldsymbol{f}_i^\star|\mathcal{G}^\star,\{\mathcal{G},\boldsymbol{Y}\})Q_\theta(\mathcal{G})\,d\mathcal{G} \tag{41}$$

$$\approx \frac{1}{K}\sum_{k=1}^{K} p(\boldsymbol{f}_i^\star|\mathcal{G}^\star,\{\mathcal{G}_k,\boldsymbol{Y}\}) \tag{42}$$

where each $\mathcal{G}_k$ is a set of $M$ latents (one for each of the $M$ conditions in the data $\boldsymbol{Y}$) sampled from the variational posterior $Q_\theta(\mathcal{G})$. The standard deviation around the mean tuning curves in all figures are estimated from 1000 independent samples from this posterior, with each draw involving the following two steps: (i) draw a sample $\mathcal{G}_k$ from $Q_\theta$ and (ii) conditioned on this sample, draw from the predictive distribution $p(\boldsymbol{f}_i^\star|\mathcal{G}^\star,\{\mathcal{G}_k,\boldsymbol{Y}\})$. Together, these two steps correspond to a single draw from the posterior. Note that we make a variational sparse GP approximation (Section 2.2.2) and therefore approximate the predictive distribution $p(\boldsymbol{f}_i^\star|\mathcal{G}^\star,\{\mathcal{G}_k,\boldsymbol{Y}\})$ as described in Titsias (2009).

## F  Alignment for visualization

The mGPLVM solutions for non-Euclidean spaces are degenerate because the ELBO depends on the sampled latents through (i) their uniform prior density, (ii) their entropy, and (iii) the GP marginal likelihood, and all three quantities are invariant to transformations that preserve pairwise distances. For example, the application of a common group element $g$ to *all* the variational means leaves pairwise distances unaffected and therefore does not affect the ELBO. Additionally, pairwise distances are invariant to reflections along any axis of the coordinate system we have chosen to represent each group. Therefore, to plot comparisons between true and fitted latents, we use numerical optimization to find a single distance-preserving transformation that minimizes the average geodesic distance between the variational means $\{g_j^\mu\}$ and the true latents $\{g_j\}$.

For the $n$-dimensional torus (Figures 2 and 3) which we parameterize as

$$\boldsymbol{g} \in \{(g_1,\cdots,g_n);\forall k: g_k \in [0,2\pi]\},$$

the distance metric depends on $\cos(g_k - g_k')$ and is invariant to any translation and reflection of all latents along each dimension

$$g_k \to (\alpha_k g_k + \beta_k) \mod 2\pi$$

where $\alpha_k \in \{1,-1\}$ and $\beta_k \in [0,2\pi]$. We optimize discretely over the $\{\alpha_k\}$ by trying every possible combination, and continuously over $\beta_k$ for each combination of $\{\alpha_k\}$.

In the case of $S^2$, $S^3$ and $SO(3)$ (Figures 3 and 6), the distance metrics are invariant to unitary transformations $\boldsymbol{g} \to \boldsymbol{Rg}$ where $\boldsymbol{RR}^T = \boldsymbol{R}^T\boldsymbol{R} = \boldsymbol{I}$ for the parameterizations used in this work. For visualization of these groups, we align the inferred latents with the true latents by optimizing over $\boldsymbol{R}$ on the manifold of orthogonal matrices.

## G  Automatic relevance determination

As we mention in Section 4, it is possible to exploit automatic relevance determination (ARD) for automatic selection of the dimensionality of groups with additive distance metrics such as the $T^n$-distance in Equation 29. While we have not investigated this in detail, we illustrate the idea here on a simple example. We consider the same synthetic data as in Figure 2 and fit a $T^2$-mGPLVM with a kernel on $T^2$ that has separate lengthscales $\ell_1$ and $\ell_2$ for each dimension:

$$k_{T_{\text{ARD}}^2}(\boldsymbol{g},\boldsymbol{g}') = \alpha^2 \exp\left(\frac{\cos(g_1 - g_1') - 1}{\ell_1^2}\right)\exp\left(\frac{\cos(g_2 - g_2') - 1}{\ell_2^2}\right). \tag{43}$$

Figure 7: **Automatic relevance determination (ARD) in $T^2$-mGPLVM.** A $T^2$ model with ARD was fitted to the $T^1$ data in Figure 2. **(a)** Length scales along each of the two dimensions for each neuron. **(b)** Posterior variational distributions. Shading indicates $\pm 1$ s.t.d. around the posterior mean in each dimension. **(c)** Variational mean plotted against the true latent state for each dimension.

Additionally, we assume the variational distribution to factorize across latent dimensions:

$$q_{\theta_j}(\cdot) = q_{\theta_j^1}(\cdot)\, q_{\theta_j^2}(\cdot), \tag{44}$$

such that their entropies add up to the total entropy:

$$H(q_{\theta_j}) = H(q_{\theta_j^1}) + H(q_{\theta_j^2}). \tag{45}$$

This corresponds to assuming that each variational covariance matrix $\boldsymbol{\Sigma}_j$ (Section 2.2.1) is diagonal.

When fitting this model, we find that one length parameter goes to large values while the other remains on the order of the size of the space (Figure 7a; note that $d_{T^1} \in [0, 4]$). This indicates that neurons are only tuned to one of the two torus dimensions. Additionally, posterior variances become very large in the non-contributing dimension, i.e. the data does not contain the other angular dimension (Figure 7b). This further indicates that the model has effectively shrunk from a 2-torus to a single circle. We note that the entropy of the factor in the variational posterior that corresponds to the discarded dimension becomes $\log 2\pi$ as the variance goes to infinity in this direction. This exactly offsets the increased complexity penalty of the prior for $T^2$ compared to $T^1$, such that the two models have the same ELBO. The model thus reduces to a $T^1$ model, demonstrating how ARD can be exploited to automatically infer the dimensionality of the latent space.

## H    Direct products of Lie groups

Here, we elaborate slightly on the extension of mGPLVM to direct products of Lie groups, briefly mentioned in the discussion (Section 4). Assuming additive distance metrics and factorizable variational distributions, direct product kernels become multiplicative and entropies become additive – very much as in our illustration of ARD in Appendix G. That is, for a group product $\mathcal{M} = \mathcal{M}_1 \times \ldots \times \mathcal{M}_L$, we can write

$$k^{\mathcal{M}}(g, g') = \prod_l k^{\mathcal{M}_l}(g, g'), \tag{46}$$

$$H(q_{\theta_j}^{\mathcal{M}}) = \sum_l H(q_{\theta_j}^{\mathcal{M}_l}). \tag{47}$$

As a simple example, we consider a $(T^1 \times \mathbb{R}^1)$-mGPLVM which we fit to the *Drosophila* data from Section 3.2. Here we find that the $T^1$ dimension of the group product, which we denote by $\theta^{(T^1 \times \mathbb{R}^1)}$, captures the angular component of the data since it is very strongly correlated with the latent state $\theta^{T^1}$ inferred by the simpler $T^1$-mGPLVM (Figure 8a). It is somewhat harder to predict what features of the data will be captured by the $\mathbb{R}^1$ dimension $x^{(T^1 \times \mathbb{R}^1)}$ of the $(T^1 \times \mathbb{R}^1)$-mGPLVM, but we hypothesize that it might capture a global temporal modulation of the neural activity. We therefore plot the mean instantaneous activity $\bar{y}$ across neurons against $x^{(T^1 \times \mathbb{R}^1)}$ and find that these quantities

Figure 8: $(T^1 \times \mathbb{R}^1)$-**mGPLVM.** (a) Latent states inferred by $T^1$-mGPLVM (Figure 4a) against the periodic coordinate of a $(T^1 \times \mathbb{R}^1)$-mGPLVM fitted to the *Drosophila* data. (b) Momentary average population activity $\bar{y}_t$ against the scalar Euclidean component of the $(T^1 \times \mathbb{R}^1)$ latent representation.

are indeed positively correlated (Figure 8b). This exemplifies how an mGPLVM on a direct product of groups can capture qualitatively different components of the data by combining representations with different topologies.

This direct product model is very closely related to the ARD model in Appendix G, and the two can also be combined in a direct product of ARD kernels. For example, we can imagine constructing a $(T^n \times \mathbb{R}^n)$ direct product ARD kernel which automatically selects the appropriate number of both periodic and scalar dimensions that best, and most parsimoniously, explains the data.

# I   Implementation

**Scaling**   As mentioned in Section 2.2.2, approximating the GP likelihood term $\mathbb{E}_{Q_\theta}[\log p(\boldsymbol{Y}|\{g_j\})]$ in the mGPLVM ELBO scales as $\mathcal{O}(m^2 MNK)$ with $m$ inducing points, $M$ latent states, $N$ neurons, and $K$ Monte Carlo samples. Estimating the entropy term is $\mathcal{O}(MKd)$ for a $d$-dimensional Euclidean latent space, $\mathcal{O}(MK(2k_{max}+1)^d)$ for a d-dimensional torus, and $\mathcal{O}(MK(2k_{max}+1))$ for $SO(3)$ and $S^3$, where $k_{max}$ is the maximum value of $k$ used in Equation 8. For all manifolds considered in this work, we can compute a closed-form $\mathrm{Exp}(\cdot)$ while for general matrix Lie groups, approximating Exp as a power series is $\mathcal{O}(d^3)$ (Falorsi et al., 2019), further increasing the complexity of mGPLVM for such groups.

For our manifolds of interest, computing the likelihood term tends to be the main computational bottleneck, although the entropy term can become prohibitive for high-dimensional periodic latents (Rezende et al., 2020). When computing $\mathbb{E}_{Q_\theta}[\log p(\boldsymbol{Y}|\{g_j\})]$, most of the complexity is due to inverting $NK$ matrices of size $(Mm^2) \times (Mm^2)$, which can be performed in parallel for each Monte Carlo sample and neuron. Using PyTorch for parallelization across neurons and MC samples, we can train $T^1$-mGPLVM with $N = 300$ and $M = 1000$ in $\sim 100$ seconds on an NVIDIA GeForce RTX 2080 GPU with 8GB RAM.

**Initialization**   For all simulations, we initialized the system with variational means at the identity element of the manifold, but with large variational variances to reflect the lack of prior information about the true latent states. Inducing points were initialized according to the prior on each manifold (Equation 1). To avoid variational distributions collapsing to the uniform distribution early during learning, we ran a preliminary 'warm up' optimization phase during which some of the parameters were held fixed. Specifically, we fixed the variational covariance matrices as well as the kernel variance parameters ($\alpha$ in Equation 13), and prioritized a better data fit by setting the entropy term to zero in Equation 5. Learning proceeded as normal thereafter.

**Entropy approximation**   When evaluating Equation 8, we used values of $k_{max} = 3$ for the tori and $S^3$ as in Falorsi et al. (2019) and $k_{max} = 5$ for $SO(3)$ since the sum takes steps of $\pi$ instead of $2\pi$. In theory, the finite $k_{max}$ can lead to an overestimation of the ELBO for large variational uncertainties, as $\tilde{q}$ is systematically underestimated, leading to overestimation of the entropy. To mitigate this, we capped the approximate entropy for non-Euclidean manifolds at the maximum entropy corresponding to a uniform distribution on the manifold.