[Reviews · NeurIPS 2020]

Review 1

Summary and Contributions: The authors propose a novel extension to the Bayesian Gaussian Process Latent Variable (GPLVM) model, where the latent variable is confined to non-Euclidean geometries. Performing approximate inference in this model is met with challenges due to intractabilities in evaluating the variational lower bound. To address these, the authors build on previous work using the reparameterization trick for distributions defined on Lie groups. Using specific topologies, the authors demonstrate improved performance on synthetic and real data compared to using a GPLVM with a Euclidean latent space.

Strengths: The work represents an interesting extension to GPLVM. It is likely to have broad applications and should be relevant to the NeurIPS community. The results demonstrate that the ability to incorporate information about non-Euclidean geometry into the model provides an improvement over classic Euclidean approaches.

Weaknesses: There is no discussion of scalability/computational complexity. It would be important to clearly state how the algorithm scales in terms of the latent dimensionality, inducing point numbers, number of Monte Carlo samples, etc. Furthermore, for all results the generative topology has only been compared to the Euclidean alternative. It would be interesting if it could be demonstrated that the correct topology can be selected via cross-validation from a range of options. This would be important for neuroscience applications, where the variables that are encoded (and hence the appropriate prior, kernel, etc) are unknown a priori. In the rebuttal, the authors address both of these concerns. A discussion of computational complexity has been added, which I do think is valuable. Furthermore, the authors demonstrate that the correct topology can be inferred by cross-validation on synthetic datasets.

Correctness: As far as I can tell the methodology of the paper are correct. In line 46 the authors mention that the inference is scalable, yet there is no complexity analysis of the algorithm to demonstrate this. The complexity analysis and a more thorough discussion of scalability has been added in the rebuttal thus addressing this concern.

Clarity: The paper is well written and well structured. The exposition is clear with additional details in the appendix.

Relation to Prior Work: The authors discuss clearly how the extension differs from the classic GPLVM. The authors mention normalizing flows only in passing as a footnote. A more extensive discussion of differences and benefits over non-GPLVM approaches to similar problems would be useful. In the rebuttal, the authors have stated that they will include this in a revised manuscript, which addresses my concern here.

Reproducibility: Yes

Additional Feedback: It would be helpful to provide a few more details on the approximation in line 116. What are "the first few dominant terms" exactly? What does the approximation mean for the variational lower bound? Given that the main motivation of the work lies in neuroscience, it might also be interesting to discuss the possibility of non-Gaussian observation models such as in [37].


Review 2

Summary and Contributions: This paper presents a latent variable model for uncovering various neural transformations and representations, particularly those that are better encoded in non-Euclidean spaces. The main contribution of this paper is an extension to a Gaussian process latent variable model that allows for latent spaces such as tori, spheres and SO(3). From the modelling perspective, the paper combines a GPLVM with the reparameterisation trick for Lie groups of Falorsi et al. The paper focuses on applications to neural data.

Strengths: The motivation for the work is sound and the tool developed in this paper may be of interest to the wider community, especially those working on unsupervised learning in neural data where latent space representations may help uncover some underlying traits of some neural activity. The authors provide some biological interpretation for the latent space and for the GP mappings (which they refer to as "tuning curves"). The main modelling contribution of the paper is given in Sec. 2.3, which is well written and provides some intuition of what the topological constraints imply in terms of changes to the distributions.

Weaknesses: The proposed approach seems like a fairly straight-forward combination of the well-established GP latent variable model and the recently proposed reparameterisation trick for distributions on Lie groups. As such, the technical contribution is not huge. It is somewhat harder to access the quality of the contribution in terms of the applications, though it seems to work well on toy examples.

Correctness: As far as I can tell, the methodology is sound.

Clarity: The paper is well-written, easy to follow and quite constructive.

Relation to Prior Work: In general, there's a reasonable discussion of previous work. In addition, some of the same ideas on topologically constraining the latent space were explored in [1]. While the treatment in [1] is less formal and they do not perform variational inference (and use MAP estimates instead), it might still be interesting to compare the two approach. [1] R. Urtasun, et at. Topologically-Constrained Latent Variable Models (2008) ICML.

Reproducibility: Yes

Additional Feedback: I think it would be good if you moved some of the comments that are currently in the notes for the implementation to the main part of the paper (or at least to the appendix). For example this: "For all calculations, a 'burn-in' period was used where variational covariances were fixed and no entropy was included in the ELBO. This was found to be particularly important in periodic spaces to avoid an early collapse of the variational distribution." My understanding is that many of these and related models are quite hard to train and there is no shame in admitting that. If there are some tools or tricks that help with the optimisation, it should be made explicit to aid the use of this and related models in the future. Also, some of the tools and tricks might have an interpretation for why they work (as you indicated in the comments of the implementation) and that is worth including and discussing in the paper. It would be good to include a discussion of the computation complexity as well. Are there any other issues that might occur when scaling this method to larger datasets? Furthermore, it seems to me that the initialisation (or the priors) are very important for training this model as there are infinitely many equally good solutions (due to rotational symmetries), as you mention in line 182. Do you have any further comments on that? Have you considered applying the reparameterisation trick to other latent space models? Or rather, why did you choose to use GPLVM, in particular. Would it be possible to use PPCA instead (I assume not due to the fact that the mappings in PPCA are linear)? You mention in line 180 that for practical reasons you use the same kernel parameters for all neurons. Are there any situations where you would want these parameters to be different? My understanding is that standard GPLVM the parameters for these GPs are always the same for all features (neurons in your case). ---------------------------AFTER REBUTTAL------------------------------------------- I would like to thank the authors for such a thorough and carefully written rebuttal, it has definitely helped the reviewers reach a consensus.


Review 3

Summary and Contributions: The paper propose an extension of the Gaussian Process Latent Variable Model (GP-LVM) where the latent space is a pre-defined Lie group (supplements loosen this assumption to be a Riemannian manifold). This is achieved through a combination of existing techniques. The key contribution appear to be that the model is particularly tailored to model neurological data, where we are often interested in an explicit model of things like head orientation. == After the rebuttal == The authors provided a very strong rebuttal, and after discussions with my fellow reviewers I have raised my score. It seems the neuroscience aspect of the work carries value (which I am unable to evaluate) to the community. As a last note: the 'related work' table in the rebuttal was nice for the rebuttal, but I do not think such an explicit table is needed in the paper.

Strengths: The paper is well-written, easy to follow, and generally describe a sensible contribution. The technique is a combination of existing ones, but the end-result is a novel model / technique. The proposed model is tailored for neurological data, and the arguments behind the model appear sensible. This is, however, beyond my own expertise, so I cannot comment on the significance of this contribution.

Weaknesses: From my perspective, I see two concerns with the paper: (1) The model is realized using a set of existing tools, such that the methodological contribution is somewhat limited. This is not a problem per se: if the resulting model is useful, then that is a valuable contribution. I am, however, not qualified to judge the usefulness of the model, as I am not well-versed in the neuro-science domain. (2) Quite a bit of highly related work is currently missing from the paper. I am unaware of related work that diminished the contribution of the paper, but it is something I would like to see improved upon. More on this below.

Correctness: As far as I can tell, the proposed work is correct. One somewhat misleading part of the paper is the way positive definite kernels are defined over Riemannian latent spaces. This is achieved by disregarding the geometry (but not topology) of the latent space. As the authors are predominantly interested in topology, this is acceptable, but a more direct communication of these issues would have been nice.

Clarity: The paper is very well written and is quite easy to follow.

Relation to Prior Work: Quite a few relevant papers are missing from the paper. From what I can tell, none of the papers below diminishes the contribution of the present paper, but I think previous work could be much better acknowledged: (*) "Geodesic Exponential Kernels: When Curvature and Linearity Conflict". Feragen et al., CVPR 2015. This paper discuss the difficulties with designing positive definite kernels over curved spaces. It shows that reference 14 in the present paper is largely incorrect. (*) "Probabilistic Riemannian submanifold learning with wrapped Gaussian process latent variable models". Mallasto et al., AISTATS 2019. This paper considers the case where data resides on a Riemannian manifold, and the latent space is Euclidean. This is the opposite case of what is considered in the present paper. (*) "Metrics for Probabilistic Geometries", Tosi et al., UAI 2014. This paper consideres a GP-LVM with a learned Riemannian latent space. In some sense this is more general than what is being proposed here, with the (important) exception that no priors are considered over the latent space. (*) "Tracking People on a Torus" Elgammel & Lee, PAMI 2009. This paper considers a GP-LVM like model where the latent space is torus. No priors over the latent space are considered and the application domain is significantly different. (*) "Hyperspherical Variational Auto-Encoders", Davidson et al. 2018, https://arxiv.org/abs/1804.00891. This paper is one of many that consider latent spaces over predefined manifolds in a variational autoencoder. The main difference to the present paper is that one paper is in a VAE setting while the other is in a GP-LVM setting. (*) "Latent Space Oddity: on the Curvature of Deep Generative Models". Arvanitidis et al. ICLR 2018. This paper also considers VAEs with Riemannian latent spaces, but here the geometry is learned rather than pre-specified. Here a prior over the latent space is not considered.

Reproducibility: Yes

Additional Feedback: I am not qualified to evaluate the neuro-science aspect of the proposed paper. I hope another reviewer is qualified in this direction.

[Author Response · NeurIPS 2020]

We thank the reviewers for their suggestions and for appreciating the clarity of the paper (**R1**, **R2**, **R3**), novelty of
mGPLVM (**R1**, **R3**), and its relevance to NeurIPS (**R1**, **R2**). To address their main concerns, we now apply mGPLVM
to two additional datasets, add extensive comparisons to prior work, and incorporate their minor comments.

**Applications (R1, R2, R3)**   We developed mGPLVM as a novel combination of ML techniques with the specific
aim of addressing open problems in systems neuroscience, where tools for analysing non-Euclidean neural codes
are currently lacking. This is important as the field sets out to unravel the neural correlates of computations on
non-Euclidean manifolds, such as motor control [35], path integration [Burak *PLoS Comput Biol* 2009], and mental
processing of object transformations [28]. To help our reviewers appreciate the expected impact of mGPLVM in
neuroscience (**R3**), we have extended our demonstration of its uses in two directions.
**R1**, **R3**: First, we show that mGPLVM can distinguish between non-Euclidean latent topologies (panel (**a**); three-fold
comparison of $T^2$, $SO(3)$ and $S^3$). Such comparisons could e.g. be used to resolve the neural encoding of 3D heading in
the bat [10, Rouault *Cosyne* 2017]. Moreover, Bayesian model comparison enables exploratory (instead of confirmatory)
analysis of neural population codes. In this case, revealing the topology of an unknown neural representation strongly
constrains the identity of the latent variable(s) being encoded.
**R2**, **R3**: Second, we apply mGPLVM to a dataset from the mouse head direction system [Peyrache *Nat Neurosci* 2015],
revealing a clear encoding of heading. This encoding is conserved during sleep, showing that mGPLVM can be used
to infer momentary, imaginary heading representations in the absence of behaviour. Importantly, this non-toy dataset
highlights the importance of learning different kernel parameters for each neuron, in contrast to the synthetic data
of Figures 2 & 3 where neurons were homogeneously tuned by construction. In real data with heterogenous neural
populations, the learned length scales instead reveal which neurons contribute to the latent representation (panel (**b**)).

(**a**) Test log likelihood ratio for 10 synthetic datasets on $T^2$, $SO(3)$, & $S^3$, with mGPLVM fitted on each manifold (x-axis). Solid lines indicate mean across datasets. The correct topology is inferred for *all* 30 datasets. (**b**) Kernel length scales for 53 neurons in a mouse head direction circuit during sleep. Dashed line: $\ell = 4$ (maximum $d$ in the $T^1$-kernel). Insets: example neurons with low and high $\ell$.

**Related work**   **R1**, **R2** and **R3** all highlight the importance of discussing related literature on normalizing flows (**R1**),
GPLVMs with non-Euclidean features (**R2**, **R3**), linear methods (**R2**), Riemannian kernel methods (**R3**) etc. We have
now included a paragraph & table in the discussion to this effect. A subset of these comparisons is shown below, with
orange indicating the key desiderata that shaped the development of mGPLVM. Note that a tractable marginal likelihood
allows for Bayesian model selection across topologies and dimensionalities (panel (**a**); [33]).

|  | non-Eucl. latent | non-Eucl. output | latents → data | prior over $f(\cdot)$ | marginal likelihood |
|---|---|---|---|---|---|
| **mGPLVM (ours)** | ✓ |  | GP | ✓ | ✓ |
| GPFA [Yu 2009] |  |  | linear | ✓ | ✓ |
| non-Eucl. VAE [multiple] | ✓ |  | neural net |  | ✓ |
| WGPLVM [Mallasto 2019] |  | ✓ | wrapped GP | ✓ |  |
| TC-LVMs [Urtasun 2008] | (✓) |  | GP | ✓ |  |

**Scaling and implementation (R1, R2)**   mGPLVM scales as $\mathcal{O}(m^2MNK + MKC^d)$ with $m$ inducing points, $M$
latent states, $N$ neurons, $K$ Monte Carlo samples, and a $d$-dimensional latent state, for manifolds with closed-form
$\text{Exp}(\cdot)$ [9]. While the first term dominates for our manifolds of interest, the second term can be prohibitive e.g. for
high-dimensional tori [Rezende *ICML* 2020]. $C$ depends on the manifold and desired approximation accuracy of the
entropy. PyTorch allows for parallelization across neurons and MC samples, and we can train $T^1$-mGPLVM with
$N = 300$ and $M = 1000$ in 103 seconds on an NVIDIA GeForce RTX 2080 GPU with 8GB RAM. We now discuss
complexity and implementation details (**R2**), as well as the approximation in line 116 (**R1**; also discussed in ref [9]).

**Miscellaneous**   **R1** notes that an interesting extension of mGPLVM would be to incorporate a Poisson noise model
for spike count data [37], which we imagine will be possible using methods from [Hensman *JMLR* 2015].
**R2** comments on initialization, and we find this to be less important in mGPLVM than MAP-based methods since the
initial latent uncertainty can be reflected in the initial variational distributions. In fact, we obtained good results by
simply initializing all latent means at the same point on the manifold.
**R3** notes that our kernels constrain topology rather than geometry, which we now clarify in section 2.3. On a final note,
we envisage that several of the methods highlighted by **R3** could be combined with mGPLVM in future work, such as
[Feragen *CVPR* 2015] for alternative kernels, [Mallasto *AISTATS* 2019] for a non-Euclidean latent with Riemannian
observations, and [Mallasto *CVPR* 2018] to put a GP prior on the non-Euclidean latent states over time.

[Meta-Review · NeurIPS 2020]

The reviewers evaluated this paper both with respect to its usefulness from a neuroscience perspective, as well as with respect to its correctness and novelty from a machine learning perspective. After discussion, the consensus was that this method has potential to be of wide usability in neuroscience application, and that it represents a useful and clear extension of previously developed ML methods for this application. We congratulate the authors on the paper acceptance, and would ask them to modify the paper with respect to the constructive comments by the reviewers.